# The Use of *Candida pyralidae* and *Pichia kluyveri* to Control Spoilage Microorganisms of Raw Fruits Used for Beverage Production

**DOI:** 10.3390/foods8100454

**Published:** 2019-10-06

**Authors:** Maxwell Mewa-Ngongang, Heinrich W. du Plessis, Seteno Karabo Obed Ntwampe, Boredi Silas Chidi, Ucrecia Faith Hutchinson, Lukhanyo Mekuto, Neil Paul Jolly

**Affiliations:** 1PostHarvest and Agro-Processing Technologies, ARC Infruitec-Nietvoorbij (The Fruit, Vine and Wine Institute of the Agricultural Research Council), Private Bag X5026, Stellenbosch 7599, South Africa; DPlessisHe@arc.agric.za (H.W.d.P.); boredi2002@gmail.com (B.S.C.); HutchinsonU@arc.agric.za (U.F.H.); JollyN@arc.agric.za (N.P.J.); 2Bioresource Engineering Research Group (BioERG), Department of Biotechnology, Cape Peninsula University of Technology, P.O. Box 652, Cape Town 8000, South Africa; NtwampeS@cput.ac.za (S.K.O.N.); lukhayo.mekuto@gmail.com (L.M.); 3Department of Chemical Engineering, Cape Peninsula University of Technology, P.O. Box 652, Cape Town 8000, South Africa; 4Department of Chemical Engineering, University of Johannesburg, PO Box 17011, Johannesburg 2028, Gauteng, South Africa

**Keywords:** biopreservation compounds, *Candida pyralidae*, *Pichia kluyveri*, biological control, volatile organic compounds

## Abstract

Undesired fermentation of fruit-derived beverages by fungal, yeast and bacterial spoilage organisms are among the major contributors of product losses in the food industry. As an alternative to chemical preservatives, the use of *Candida pyralidae* and *Pichia kluyveri* was assessed for antimicrobial activity against several yeasts (*Dekkera bruxellensis, Dekkera anomala, Zygosaccharomyces bailii*) and fungi (*Botrytis cinerea*, *Colletotrichum acutatum* and *Rhizopus stolonifer*) associated with spoilage of fruit and fruit-derived beverages. The antagonistic properties of *C. pyralidae* and *P. kluyveri* were evaluated on cheap solidified medium (grape pomace extract) as well as on fruits (grapes and apples). Volatile organic compounds (VOCs) from *C. pyralidae* and *P. kluyveri* deemed to have antimicrobial activity were identified by gas chromatography-mass spectrometry (GC-MS). A cell suspension of *C. pyralidae* and *P. kluyveri* showed growth inhibition activity against all spoilage microorganisms studied. Direct contact and extracellular VOCs were two of the mechanisms of inhibition. Twenty-five VOCs belonging to the categories of alcohols, organic acids and esters were identified as potential sources for the biocontrol activity observed in this study. This study reports, for the first time, the ability of *C. pyralidae* to inhibit fungal growth and also for *P. kluyveri* to show growth inhibition activity against spoilage organisms (*n* = 6) in a single study.

## 1. Introduction

Fruit and fruit-derived beverages are important nutritional and economically viable commodities. They also provide vitamins and some essential minerals in the human diet. However, fruit losses caused by spoilage fungi pose numerous challenges to the agrifood industry [1,2]. Additionally, the shortened shelf-life of fruit caused by microbial spoilage is a factor that negatively affects the market value and processing of fruit. It has long been known that fungal spoilage organisms can contaminate agricultural produce before, during and after harvest, as well as during transportation and processing [3,4,5,6]. During postharvest handling, about 25% of the total harvested fruit is lost due to spoilage fungi [7,8,9,10], which is an amount of food that can feed millions of households worldwide. Spoilage organisms, i.e., *Botrytis cinerea*, *Colletotricum acutatum* and *Rhizopus stolonifer,* causes losses in more than 200 crop species [11,12]. Mycotoxins are secondary metabolites of fungi toxic to animals and humans [13]. The most important fungal genera producing mycotoxins that are found in food products are *Aspergillus*, *Fusarium*, *Alternaria* and *Penicillium* [14,15]. The components of fruit and fruit-derived beverages, i.e., sugar content, macro- and micro-nutrients, are conducive to microbial proliferation and therefore, prone to microbial spoilage. As a health risk and mode of infection, the patulin producing strain such as *Penicillium expansum* has been published as one of the most serious postharvest contaminant of apple and derived juices [13,16]. Although the presence of some mycotoxin is almost unavoidable, there are set limits, deemed safe for consumption, that are allowed on raw fruits and the derived beverages [16,17,18,19]. Even if there are safe limits for mycotoxins in fruits and the derived beverages, it is imperative to combat contamination caused by fungi in order to ensure that toxic level of contamination is avoided as this can be lethal to consumers. Other spoilage microorganisms such as *Dekkera bruxellensis*, *Dekkera anomala* and *Zygosaccharomyces bailii* are known to be responsible for spoilage and product losses in the beverage industry [20,21,22]. To minimise such losses, synthetic chemicals are currently being used for controlling microbial spoilage, but their continued use as postharvest control agents of fruit or as preservatives, raises human health concerns. As a result, the eradication of these chemicals and the development of safer alternatives is now a priority in fruit and beverage industries, i.e., agro-processing and retail industries [8,23,24,25].

As an alternative to synthetic chemicals, the antagonistic mechanisms of yeasts against fruit and beverage spoilage microorganisms have been investigated [11,26,27,28]. Several antagonistic characteristics are associated with the ability of the yeasts to rapidly reproduce on simple nutrients and to colonise surfaces, while competing for nutrients and space with the other flora [29,30], and the induction of host resistance for some yeasts [31,32,33] and the ability of yeasts to produce cell wall degrading enzymes such as laminarinases, glucanases, proteases, peroxidase and chitinases [33,34,35,36]. Other antagonistic mechanisms can be attributed to the effect of decreasing germ tube length, preventing mycelial growth and suppressing conidial germination [37], the parasitism being associated with the release of hydrolase, the oxidative stress [33] as well as the spoilage fungi and growth inhibition by diffusible volatile organic compounds (VOCs) [38,39,40,41] in synergy with a carbon dioxide enclosed environment [42]. Species of *C. pyralidae* have previously been studied as producers of killer toxins which can act against beverage spoilage organisms [43]. However, many have not yet been explored as potential biocontrol agents against fruit-spoilage organisms. Similarly, *P. kluyveri* has also been determined as an antagonistic yeast for biocontrol applications [44,45]. 

Therefore, the aim of this study was to explore whether *C. pyralidae* and *P. kluyveri* strains can control the growth of beverage spoilage yeasts and fungi contaminating fruits used for beverage production.

## 2. Materials and Methods 

### 2.1. Microorganisms and Culture Conditions 

The yeasts *C. pyralidae* (Y1117, isolated from grape must), *P. kluyveri* (Y1125, isolated from *Sclerocarya birrea* juice) and the beverage spoilage yeasts *D. bruxellensis* (ISA 1653), *D. anomala* (MSB/1) and *Z. bailii* (Y0070) were obtained from the ARC Infruitec-Nietvoorbij culture collection (Stellenbosch, South Africa). Furthermore, spoilage and disease causing fungi in the South African fruit industry, *B. cinerea*, *C. acutatum* and *R. stolonifer,* were obtained from the postharvest control laboratory at ARC Infruitec-Nietvoorbij (Stellenbosch). *C. pyralidae* and *P. kluyveri* cells were prepared by transferring a wire loop full of each culture into a test tube containing 5 mL of Yeast Peptone Dextrose (YPD) broth (Sigma Aldrich, Darmstadt, Germany) and incubated at 28 °C for 24 h. Spoilage yeast suspensions (*D. bruxellensis*, *D. anomala*, and *Z. bailii*) were prepared using YPD (Sigma Aldrich, Darmstadt, Germany) broth for 24 h at 28 °C. Cells were recovered by centrifugation at 10,000 rpm for 10 min. A haemocytometer and a microscope (400× magnification) were used to count the yeast cells and fungal spores before all mother spore solutions were diluted to the desired concentration of 10^5^ spores mL^−1^ [46].

Grape pomace from semi-industrial, conventional Chenin blanc wine grapes was obtained from the ARC Infruitec-Nietvoobij research cellar (Stellenbosch) and pressed at 200 kPa. The resultant grape pomace extract was frozen in polypropylene buckets at −10 °C. Prior to use, the grape pomace extract was thawed and diluted with sterile distilled water to the required sugar concentration (50, 100, 150 and 200 g L^−1^) and the final concentrations of the yeast assimilable nitrogen (YAN) were 0.045, 0.074, 0.136 and 0.185 g L^−1^, respectively (Arena 20XT, Thermo Fischer Scientific, Vantaa, Finland). The grape pomace extract was adjusted to pH 5, using 0.1 M NaOH prior to use. Bacteriological agar (Biolab, Merck, South Africa) was added at 10 g L^−1^ and autoclaved at 121 °C for 20 min and subsequently supplemented with 0.1 g L^−1^ chloramphenicol (Sigma-Aldrich, Darmstadt, Germany) to prevent bacterial growth. This grape pomace extract agar (GPA) was used for growth inhibition assays and grape pomace extract broth (GPB) was used as a fermentation medium.

From the 24-h-old yeast cultures, 1 mL of *C. pyralidae* and *P. kluyveri* containing broth was transferred to 150 mL sterile grape pomace extract broth (GPB) subsequent to incubation at 23 °C and agitated at 150 rpm, using a rotary shaker (LM-53OR, RKC Instrument Inc., Ohta-ku Tokyo, Japan) for 24 h [28]. For preparation of the spore solution, fungal spores of *B. cinerea*, *C. acutatum* and *R. stolonifer* were obtained from 14 day old fungal plates incubated at 20 °C. The spores were harvested by gently scrapping-off the surface with a sterile wire loop and subsequent to rinsing (*n* = 3) with sterile distilled water to obtain a 100 mL spore solution in 250 mL Schott bottles. Similarly, 5 mm diameter mycelia disks of each fungus were cut from 5-day old plates also grown on GPA at 20 °C. 

### 2.2. Effect of C. pyralidae and P. kluyveri on Spoilage Yeasts 

#### 2.2.1. Growth Inhibition Assay 

The growth inhibition assay as described by Mehlomakulu et al. [43] was used. The seeding procedures as described by Mewa-Ngongang et al. [28,43] and Mehlomakulu et al. [43] were applied. The GPA plates were seeded with 10^6^ cells mL^−1^ of *D. bruxellensis*, *D. anomala* or *Z. bailii*, respectively. *C. pyralidae* and *P. kluyveri* cells were adjusted to a concentration of 10^8^ cells mL^−1^, as described by Qin et al. [46], and 10 µL of each culture was spotted onto GPA plates seeded with *D. bruxellensis, D. anomala* or *Z. bailii*, respectively. After 72 h of incubation at 20 °C, the plates were inspected for zone of inhibition as shown by the formation of a clear zone around the yeast colonies.

#### 2.2.2. Production of Biopreservation Compounds Using Grape Pomace Extract Broth

The diluted GPB (100, 150 and 200 g L^−1^) was autoclaved for 20 min at 121 °C, cooled and was used for the production of the crude biopreservation compounds. *C. pyralidae* and *P. kluyveri* was inoculated at a concentration of 10^6^ cells mL^−1^. After initial inoculation (t = 0 h), samples (2 mL) were taken every 4 h for the duration of the experiment, which lasted for 24 h. In total, samples (*n* = 6) were taken per yeast treatment and each treatment had three replicates. Each sample was centrifuged at 10,000 rpm, and the supernatant was tested for growth inhibition activity to determine the time in the fermentation cycle at which maximum growth inhibition activity could be obtained. 

### 2.3. Effect of C. pyralidae and P. kluyveri on Fungal Growth 

#### 2.3.1. Fungal Spore Germination Assay

A radial inhibition assay was conducted using the agar plate method as described by Núñez et al. [47]. Yeast (*C. pyralidae* and *P. kluyveri*) cell suspensions (1 × 10^8^ cells mL^−1^) were prepared from the yeast culture broths and a fungal spore suspension of 10^5^ spores mL^−1^ was prepared from the mother solution. A volume (100 µL) of yeast was spread-plated on GPA and dried. Subsequently, 10 µL of 10^5^ spores mL^−1^ of each fungus was spotted at the centre of the plate with each treatment having three replicates. For the negative control plates, only 10 µL of the spore solution (10^5^ spores mL^−1^) were spotted at the centre of the GPA. The plates were then incubated at 15 °C for 7 days. The fungal radial inhibition (FRI) was calculated using the mathematical expression: FRI = (D_0_ − D_t_/D_0_) × 100, with D_0_ representing the average diameter of the fungal colony on the negative control plates and D_t_ representing the diameter of the fungal colony on the yeast treated plates [47].

#### 2.3.2. The Mouth-to-Mouth Assay and Volatile Organic Compounds (VOCs) Activity

The mouth-to-mouth assay as described by Medina-Córdova et al. [48] was used and repeated twice. Once, to study yeast inhibition and secondly, to investigate fungal growth inhibition. Two GPA plates (facing each other) were used, with the bottom plate being spread-plated with a 100 µL of the yeast suspension (10^8^ cell mL^−1^), while the top plate (cover) contained a 5 mm mycelial disk at the centre. The negative control plates were only seeded with a 5 mm diameter mycelial disk (no yeast treatment in the bottom plate). All plates (in triplicates) were sealed with laboratory film and incubated at 15 °C for 7 days. The volatile organic compounds (VOCs) inhibition activity (VOCIA) was calculated as described by the mathematical expression used for FRI (Section 2.3.1).

### 2.4. In Vivo Studies: Postharvest Efficacy of C. pyralidae and P. kluyveri in Controlling Fruit-Spoilage 

#### 2.4.1. Apple Bioassays 

Postharvest biocontrol efficacy assays were performed on “Golden Delicious” apples (*Malus domestica*). Nine treatments were investigated, each having 10 replicates. Ethanol (70% ^v^/_v_) was sprayed on the fruits and allowed to dry completely before wound infliction. Apples were uniformly wounded with a sterile cork borer (approximately 5 mm diameter and 3 mm deep). After 15 min, 15 µL of a yeast inoculum (10^8^ cell mL^−1^) was introduced into the wound and then allowed to dry for 30 min. Subsequently, 15 µL of the spore suspension (10^5^ spores mL^−1^) was introduced into the wound. Treated fruits were maintained at −0.5 °C for 4 weeks in a tightly closed container, and then stored at 20 °C for 7 days, to simulate shipping and shelf-life conditions, respectively, in a commercial setting. The containers were sterilised prior to use and were tightly closed in order to entrap the VOCs in the airspace and to observe the VOCs effect similar to the in vitro plate assay. The biocontrol efficacy was evaluated by comparing the decay diameter of the negative control to those of the treated apples using the FRI. Negative controls were prepared by inoculating fruits with 15 µL (10^5^ fungal spores mL^−1^) of *B. cinerea*, *C. acutatum* and *R. stolonifer* suspensions under similar storage conditions. Positive results were characterised by the absence of fungal development.

#### 2.4.2. Grape Bioassays

Treatments (*n* = 9), each having 10 replicates (containing 10 grape berries per jar), were evaluated for biocontrol efficacy assays which was performed on “Regal Seedless” table grapes (*Vitis vinifera*). Grapes were uniformly wounded (three wounds per spot) with a sterile needle (< 1 mm diameter per wound) and allowed to dry prior to yeast and fungal treatments. The wounded berries were sprayed with the yeast (*C. pyralidae* and *P. kluyveri*) cell suspension (10^8^ cell mL^−1^) until the dried wounds were filled with the suspension, and subsequently allowed to dry again for 30 min. The dried berries were then sprayed separately with a fungal spore suspension (10^5^ spores mL^−1^). The negative controls (10 berries each) were prepared by spraying the fungal spores suspension on the wounded berries without prior yeast treatment. All grape treatments were maintained in sterilised, sealed jars at −0.5 °C for 4 weeks, and then incubated at 20 °C for 7 days. The antagonistic properties of *C. pyralidae* and *P. kluyveri* were analysed visually by assessing the colour changes of the berries and fungal development.

### 2.5. Extraction and Gas Chromatographic Analyses of Volatile Compounds

A volume (10 mL) of the crude biopreservation samples were placed in a 20 mL headspace vial to which NaCl (30% *m*/*v*) was added. Vials were spiked with 100 µL of anisole d_8_ and 3-octanol as internal standards. SPME vials (Sigma-Aldrich, Darmstadt, Germany) were equilibrated for 5 min at 50 °C in the CTC autosampler incubator at 250 rpm. Subsequently, a 50/30 m divinylbenzene/-carboxen/-polydimethylsiloxane (DVB/CAR/PDMS) coated fibre was exposed to the sample headspace for 10 min at 50 °C. After VOCs adsorption onto the fibre, desorption of the VOCs from the fibre coating was carried out in the injection port of the gas chromatography-mass spectrometry (GC–MS) for 10 min. The fibre was inserted in a fibre conditioning station for 10 min between samples for cleaning to prevent cross- and carry-over contamination. Chromatographic separation of the VOCs was performed in a Thermo Trace 1310 gas chromatograph coupled with a Thermo TSQ 8000 mass spectrometer detector (Thermo Fisher Scientific, Vantaa, Finland). The GC–MS system was equipped with a polar DB-FFAP column (model number: J&W 122-3263), with a nominal length of 60 m, 250 μm internal diameter and 0.5 μm film thickness. Analyses were determined using helium as a carrier gas at a flow rate of 2.9 mL min^−1^. The injector temperature was maintained at 250 °C. The oven program was as follows: 350 °C for 17 min, which was then ramped to a final temperature of 240 °C at a ramping rate of 12 °C min^−1^—a temperature that was constantly maintained for 6 min. The mass selective detection (MSD) was operated in a full scan mode and both the ion source transfer line temperatures were maintained at 250 °C. Compounds were tentatively identified by blasting against the mass spectral libraries (NIST, version 2.0, Gaithersburg, MD, USA). 

### 2.6. Statistical Analysis

To determine whether there were significant differences within treatments, one-way analysis of variance (ANOVA) was performed using the general linear means procedure of SAS version 9.4 (SAS Institute Inc., Cary, NC, USA). Student’s t least significant difference (LSD) values were calculated at the 5% probability level (*p* = 0.05) to facilitate comparison between treatment means. The significant difference was calculated at the 5% level and *p* < 0.05 was considered significant for treatments. 

## 3. Results and Discussion

### 3.1. Production of Biopreservation Compounds and Their Effect on the Growth of Spoilage Yeasts 

The results in Figure 1 show that, using GPA as fermentation medium, *C. pyralidae* and *P. kluyveri* were able to secret crude extracellular compounds with growth inhibition activity against *D. bruxellensis* and *Z. bailii*, known spoilage yeasts in fruit-derived beverages. This was observed by the presence of clear zones of inhibition around the yeast colonies or agar wells. A visual observation of the clear zone of inhibition was required at this stage to verify the growth inhibition compound production and the presence of inhibition activity for the yeasts and their cell free supernatants (crude samples). Previous studies on inhibition assays were conducted using costly refined culturing media such as YPD [28,43]. However, the current study successfully managed to use a less expensive GPA for antimicrobial compound production. The findings in this work are complementary and in agreement with the research conducted by Mewa-Ngongang et al. [43]. These authors conducted a kinetic study and focused on the optimisation of biopreservation compound production using GP extract, and demonstrated that GP extract with a total sugar concentration of 150 g L^−1^ was a suitable medium for the production of antimicrobial compounds from *C. pyralidae* and *P. kluyveri* strains. 

Biopreservation compound production was shown to be sugar-dependent and the best growth inhibition was observed at a sugar concentration of 150 g L^−1^, with the biggest volumetric zone of inhibition being *C. pyralidae* against *D. bruxellensis* (Figure 1c). Similar results were observed for the *P. kluyveri* strain against *D. bruxellensis* (results not shown). Further observations indicated shorter fermentation times and lower substrate concentration can be used to reduce operational costs while attaining the desired product. Although other yeast strains are known to have growth inhibition properties against *Z. bailii* this study reports on the growth inhibition properties of *C. pyralidae* and *P. kluyveri* against *Z. bailii*. Mehlomakulu et al. [43] only reported on the growth inhibition activity of *C. pyralidae* against *D. bruxellensis*. However, this study also showed *P. kluyveri* as an antagonistic microorganism against *D. bruxellensis* (Figure 1a), with further evidence showing *C. pyralidae* having growth inhibition activities against *Z. bailii* (Figure 1b); a profile indicating *C. pyralidae* has a much broader biocontrol application. 

### 3.2. Effect of C. pyralidae and P. kluyveri Cells on Fungal Growth

A radial inhibition assay was performed in order to assess the antagonistic effect of *C. pyralidae* and *P. kluyveri* on the germination of *B. cinerea*, *C. acutatum* and *R. stolonifer* spores. A 100% inhibition against the germination of *B. cinerea*, *C. acutatum* and *R. stolonifer* was observed on the low-cost GPA plates (Figure 2A), as indicated by similar FRI-values for the three different fungi (Figure 2B). It was evident that yeasts can prevent the growth of fungi in different ways, such as the ability to outgrow the spoilage fungi and to rapidly colonise surfaces, thereby reducing fungal development. 

This is the first report on the growth inhibition properties of *C. pyralidae* and *P. kluyveri* against fruit spoilage fungi *C. acutatum* and *R. stolonifer*. However, this study also showed the potential of these yeast strains as biological control agents against *B. cinerea*. The antagonistic activity of yeast against fruit spoilage fungi was previously conducted albeit on refined culturing media [27,38,48], but not on GPA plates as carried out in this work. The usefulness of GPA as a cheap and suitable medium for antagonistic activity was demonstrated in this study.

### 3.3. Effect of Volatile Organic Compounds (VOCs) on Fungal Growth 

The influence of yeast-produced VOCs on fungal growth of fruit was investigated using a mouth to mouth assay. Compared to the negative controls, the visual (Figure 3A) and the graphical (Figure 3B) representation showed *C. pyralidae* and *P. kluyveri* producing VOCs that have an ability to completely inhibit the growth of *B. cinerea*, *C. acutatum* and *R. stolonifer*. Apart from the abovementioned cell antagonistic properties, the biocontrol mechanisms include the ability to secrete extracellular metabolites (e.g., VOCs) with growth inhibition activity against spoilage fungi [39,40,49,50]. The antagonistic effect of VOCs produced by *Debaryomyces hansenii* against *Mucor circinelloides*, *Aspergillus* sp., *Fusarium proliferatum* and *F. subglutinans* was also confirmed previously using the same mouth to mouth assay technique [48]. Contarino et al. 2019 [42] also reported the antagonistic mechanism of VOCs that was possibly coupled with the secretion of carbon dioxide in a closed packaging environment. Additionally, this study further demonstrated the potential of VOCs produced by *C. pyralidae* and *P. kluyveri* to inhibit the growth of *B. cinerea*, *C. acutatum* and *R. stolonifer*. The findings of the current study are vital in addressing fungal contamination in fruits and the derived beverages. As with many other yeast-related antagonistic studies [32,33,36,42], further investigation of the antagonistic mechanisms for *C. pyralidae* and *P. kluyveri* is recommended.

### 3.4. Postharvest Control Efficacy of C. pyralidae and P. kluyveri on Fungal Growth

#### 3.4.1. Apple Bioassay

The evaluation of the efficacy of yeasts in preventing fungal spoilage of apples using bioassays showed a considerable decay reduction (Figure 4). For the purpose of visual representation, only three apples per treatment were selected as representatives (Figure 4). The apple bioassay demonstrated the ability of *C. pyralidae* and *P. kluyveri* to control spoilage caused by *B. cinerea*, *C. acutatum* and *R. stolonifer*; however, the inhibition responses were yeast and fungal species-dependent. Near complete (100%) fungal growth inhibition was observed when both *C. pyralidae* and *P. kluyveri* were tested against *C. acutatum. C. pyralidae*, albeit only showing a 43 and 52% growth inhibition of *B. cinerea* and *R. stolonifer*, respectively, while *P. kluyveri* revealed a 38 and 22% growth inhibition of *B. cinerea* and *R. stolonifer*. Fluctuating in vivo postharvest control responses have been reported for other species including *Sporidiobolus pararoseus* [51], *Saccharomyces cerevisiae, Wickerhamomyces anomalus, Metschnikowia pulcherrima* and *Aureobasidium pullulans* [52] and *Hanseniaspora uvarum* [46]. Although *C. pyralidae* and *P. kluyveri* showed greater biocontrol potential on plate assays, the apple bioassay showed a significant decay reduction (Figure 4). These observations on apples could be of great importance in industry as these biocontrol yeasts can be further tested on patulin producing strains such as *Penicillium expansum,* which is currently a challenge in the beverage industry [16,53].

#### 3.4.2. Grape Bioassays

The grape bioassay under a closed airspace was carried out in order to assess the efficacy of the antagonistic effects of VOCs produced by *C. pyralidae* and *P. kluyveri* on table grapes, and to verify the results achieved during the in vitro assays (Figure 3). Unlike apple bioassays, grape spoilage is usually measured per bunch and not as individual berries. Therefore, the radial inhibition tests were deemed unnecessary. Interestingly, a 100% inhibition of *B. cinerea*, *C. acutatum* and *R. stolonifer* growth was observed in vivo (Figure 5). Based on these observations, the biocontrol effect (in vivo) of VOCs from *C. pyralidae* and *P. kluyveri* was confirmed by 100% inhibition or 0% decay on grapes (sealed jar settings). This showed that the VOCs played a vital role in preventing fungal growth. Undoubtedly, the complete inhibition of fungal growth makes *C. pyralidae* and *P. kluyveri* suitable candidates for postharvest control and confirms the yeast’s ability to produce VOCs with antimicrobial properties in sealed/tightly packaged fruit [51,54].

### 3.5. Identification of the VOCs Produced by C. pyralidae and P. kluyveri

VOCs (*n* = 25) were produced by each of the yeast and were identified and similar VOCs profiles were observed in both biocontrol yeast supernatants. These VOCs were isobutyl acetate, isobutanol, ethyl acetate, isoamyl acetate, limonene, isoamyl alcohol, ethyl caproate, hexyl acetate, acetoin, 4-hexen-1-ol, acetate, hexanol, nonanal, ethyl caprylate, acetic acid, trans-1-phenyl-1-butene, furfuryl acetate, 2-methyl-3-thiolanone, 4-methylbenzaldehyde, isobutyric acid, 3-(methylthio) propylacetate, ethyl dec-9-enoate, phenethyl acetate, 2,5-dimethylbenzaldehyde and 2-phenylethanol. Most of the identified compounds were esters, alcohols and fatty acids that are safely used in the food, beverages, pharmaceutical and the cosmetic industries [23]. In the context of natural biological control, the antagonistic action mechanisms of some yeasts have indeed been linked to the production of some VOCs from the category of alcohols, organic acids and esters including some of the identified VOCs are currently used as biocontrol agents in the food and beverage industries [38,39,40]. Growth inhibition properties of 2-phenylethanol [50,55], ethyl acetate [39] and acetic acid [40] have also been reported. Growth inhibition of spoilage fungi by VOCs have been linked to preventing mycelial growth and suppressing conidial germination [37].

Although the antagonistic effect of *C. pyralidae* and *P. kluyveri* is also related to the production of VOCs, it is not yet clear which of the aforementioned compounds, or combinations thereof, may be responsible for the growth inhibition activity. Hence, more research about these compounds still needs to be conducted.

## 4. Conclusions

The crude samples collected from fermentation broth inoculated with *C. pyralidae* or *P. kluyveri* showed can be used to control fruit derived beverage spoilage organisms *D. bruxellensis*, *D. anomala* and *Z. bailii*. Additionally, the cell suspensions of *C. pyralidae* and *P. kluyveri* showed growth inhibition properties against fruit spoilage fungi *B. cinerea*, *C. acutatum* and *R. stolonifera*, which is an important achievement for the alleviation of spoilage and also the mycotoxin burden faced by fruits-derived beverage producers. This study also showed that grape pomace extract can be used as a cost-effective renewable bioresource for the growth of biocontrol yeasts and production of biopreservation compounds. The potential of VOCs as a mechanism of inhibition was also shown. Future studies should investigate the mechanisms of inhibition of VOCs. Additional research is needed to determine the minimum inhibitory concentrations of individual VOCs as well as the evaluation of antimicrobial activity of pure and/or combinations of VOCs. Given the risks posed by patulin producing strain of *Penicillium* sp. on pome and stone fruits, further research should also focus on the efficacy assessments, the regulatory implications and food-safety aspects associated with the studied biological control yeasts.

## Figures and Tables

**Figure 1 foods-08-00454-f001:**
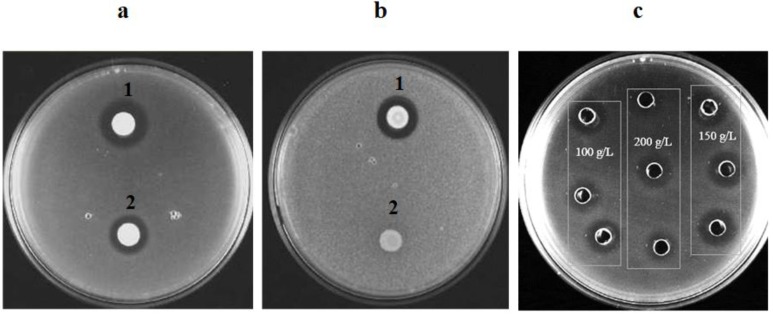
(**a**) Antagonistic activity of *C. pyralidae* (1) and *P. kluyveri* (2) against *D. bruxellensis*. (**b**) Antagonistic activity of *C. pyralidae* (1) and *P. kluyveri* (2) against *Z. bailii*. (**c**) Depiction of the inhibition activity of the biopreservation compounds produced by the antagonistic yeasts *C. pyralidae* against *D. bruxellensis* after 24 h of fermentation in grape pomace with varying sugar concentration (100, 150 and 200 g L^−1^).

**Figure 2 foods-08-00454-f002:**
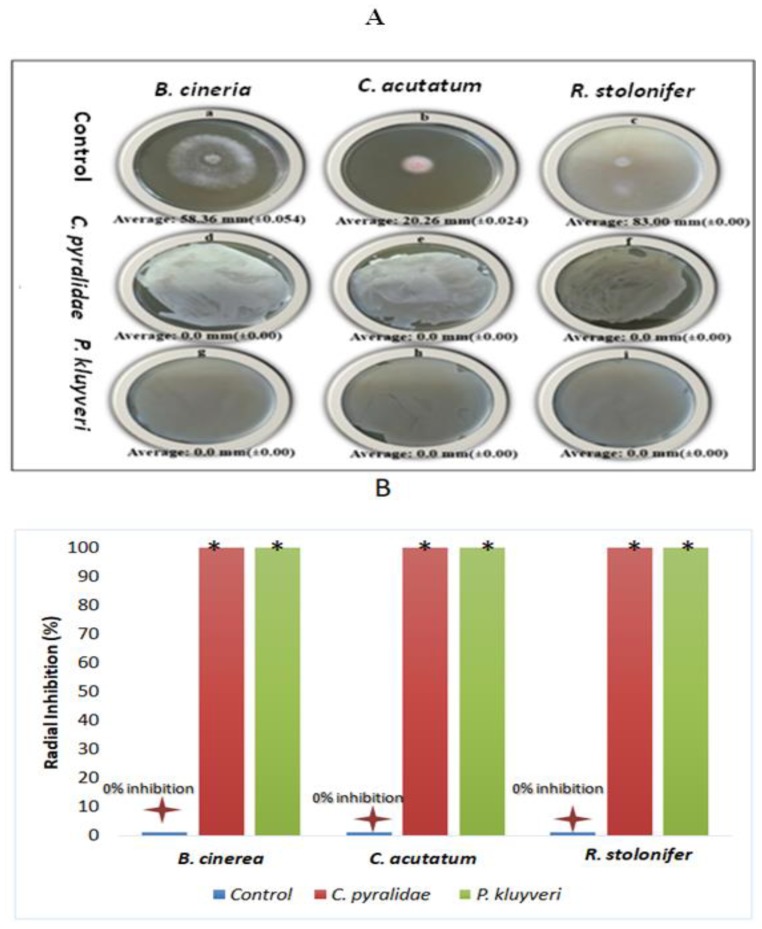
The visual (**A**) and graphical (**B**) representation the antagonistic effect of *C. pyralidae* cells on *B. cinerea* (A: d), *C. acutatum* (A: e) and *R. stolonifer* (A: f), and the antagonistic effects of *Pichia kluyveri* cells on the growth of *B. cinerea* (A: g), *C. acutatum* (A: h) and *R. stolonifer* (A: i). The negative controls are displayed as *B. cinerea* (A; a), *C. acutatum* (A: b) and *R. stolonifera* (A: c). Values are averages of three replicates per treatment with the standard deviation given in brackets. The asterisk (*) indicates values that differ significantly from the control (*p* < 0.05).

**Figure 3 foods-08-00454-f003:**
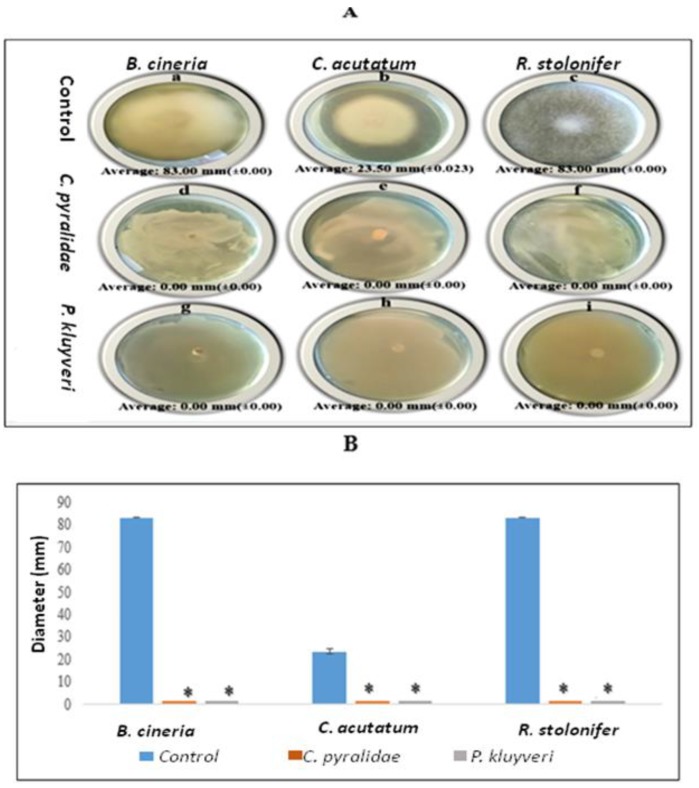
The visual (**A**) and graphical (**B**) representation the antagonistic effect of volatiles produced by *C. pyralidae* against *B. cinerea* (A: d), *C. acutatum* (A: e) and *R. stolonifer* (A: f), and the antagonistic effects of *Pichia kluyveri* against the growth of *B. cinerea* (A: d), *C. acutatum* (A: e) and *R. stolonifer* (A: f). The negative controls are displayed as *B. cinerea* (A: a), *C. acutatum* (A: b) and *R. stolonifera* (A: c). Values are averages of three replicates with the standard deviation given brackets. The asterisk (*) indicates values that differ significantly from the control (*p* < 0.05).

**Figure 4 foods-08-00454-f004:**
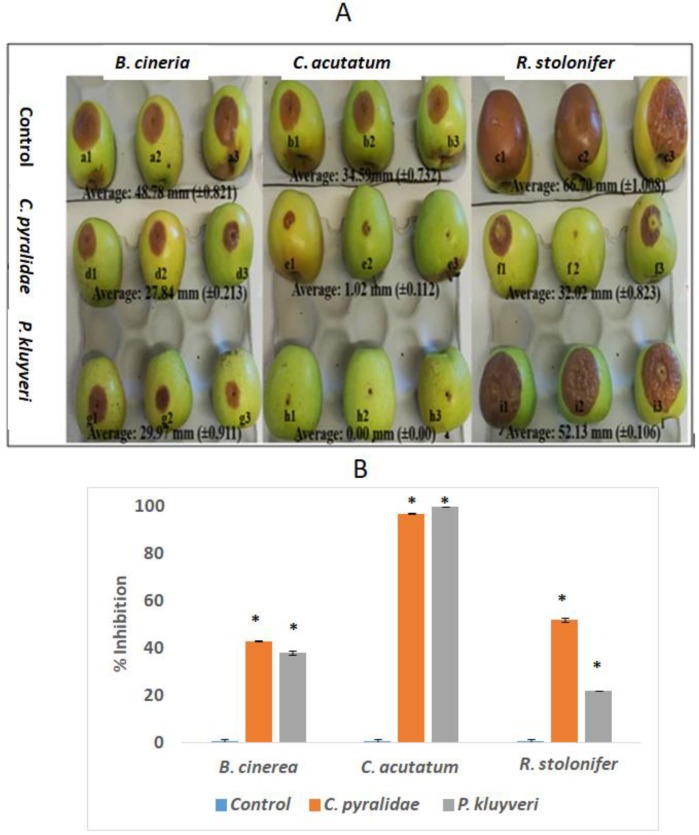
The visual (**A**) and graphical (**B**) representation of the apple bioassays for *C. pyralidae* against the growth of *B. cinerea* (A: d1–d3), *C. acutatum* (A: e1–e3) and *R. stolonifer* (A: f1–f3), and for *P. kluyveri* against the growth of *B. cinerea* (A: g1–g3), *C. acutatum* (A: h1–h3) and *R. stolonifer* (A: i1–i3). The negative controls are displayed as *B. cinerea* (A: a1–a3), *C. acutatum* (A: b1–b3) and *R. stolonifera* (A: c1–c3). Values are averages of 10 replicates per treatment with the standard deviation shown in brackets. The asterisk (*) indicates values that differ significantly from the control (*p* < 0.05).

**Figure 5 foods-08-00454-f005:**
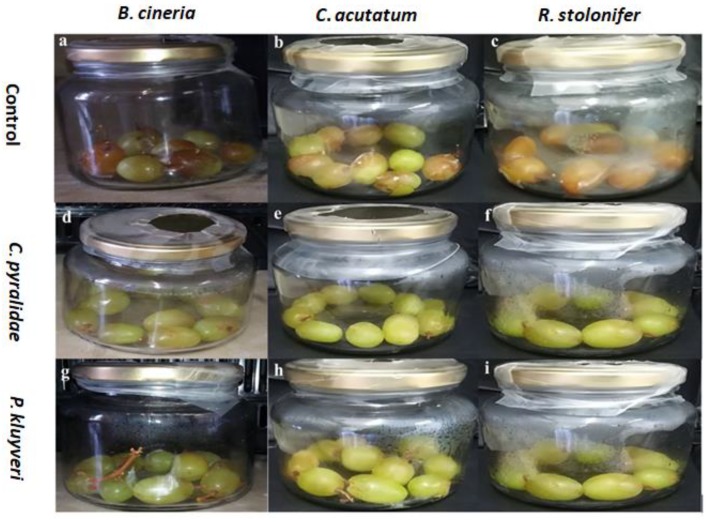
The antagonistic effect of volatiles produced by *C. pyralidae* against the growth of *Botrytis cinerea* (**d**), *C. acutatum* (**e**) and *R. stolonifer* (**f**), and the antagonistic effect of volatiles produced by *Pichia kluyveri* on *B. cinerea* (**g**), *C. acutatum* (**h**) and *R. stolonifer* in sealed jars. Negative controls are shown as *B. cinerea* (**a**), *C. acutatum* (**b**) and *R. stolonifer* (**c**). Ten replicates consisting of 10 grape berries per jar were tested. All berries in the negative control jars were spoiled (0% inhibition ± 0.00) while the treated jars were not spoiled (100% inhibition ± 0.00).

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
