# Peer review of "The Use of Candida pyralidae and Pichia kluyveri to Control Spoilage Microorganisms of Raw Fruits Used for Beverage Production"

_foods, 2019, doi:10.3390/foods8100454_

Round 1

Reviewer 1 Report

Manuscript number: 582883

Title: The Use of Candida pyralidae and Pichia kluyveri for  Control of Beverage and Fruit Spoilage Organisms

Major comments:

The manuscript titled “The Use of Candida pyralidae and Pichia kluyveri for  Control of Beverage and Fruit Spoilage Organisms” describes the use of antagonistic yeasts as biocontrol agents against food spoilage microorganisms such as fruit spoiling yeasts and fungi contaminating fruits. The manuscript has been sent for publication under the special issue “Microbial spoilage of beverages”. The authors describe the antimicrobial activity of these yeasts against fungi to control post-harvest spoilage of fresh fruits.

1 In my opinion, the objective of the work could be revised, matching with the title of the special issue. In particular, the objective could be the control of beverage spoilage microorganisms (yeasts) and fungi contaminating fruits used to produce beverages (fruit juices, smoothies) such as apple and grape described in the work. The term post-harvest and fungal pathogens could be deleted throughout the manuscript and replaced with “spoilage”, “spoilage fungi” of fruits “used for beverage production/processing”.

The most appropriate title could be “The Use of Candida pyralidae and Pichia kluyveri to Control Spoilage Microorganisms of Raw Fruits Used for Beverage Production”

2 By a scientific point of view, the results are interesting for a broad audience since different target microorganisms are considered and different modes of action of Candida pyralidae and Pichia kluyveri such as killer toxin production, VOCs release, cell to cell contact and so on could be involved in the antimicrobial activity of these yeasts. The production of crude extracts using agro-food by-products such grape pomaces is interesting and matches with current literature (as recently reported for the production of killer toxins by recombinant yeasts using other food processing by-products). However, the conclusions should reflect the results obtained. For example, the sentence “Additionally, the cell suspensions of C. pyralidae and P. kluyveri showed growth inhibition properties against fungal growth (B. cinerea, C. acutatum and R. stolonifer) through a VOCs mechanism” is correct only for in vitro assays but not for fruit challenge-tests. Indeed, the authors did not demonstrate that VOCs produced by yeasts are essentials for the antimicrobial activity detected on fruits. VOCs were detected on inoculum broths prepared from GP broth and not in the packaging headspace above the fruits; moreover, the proposed mechanism of action should require additional work related to evaluation of the antimicrobial activity of pure and/or combinations of VOCs and the definition of Minimum inhibitory concentrations of individual VOCs.

In addition, inoculum procedure adopted (yeasts and fungi were simultaneously inoculated into wounded fruits) did not exclude other direct contact mechanisms such as the production and release of antifungal compounds by yeasts (antimicrobial peptides or other compounds inhibiting spore germination).

For these reasons, I strongly suggest to be careful throughout the manuscript in the association between the antimicrobial activity detected and the mechanisms of action. They could be only suggested as “potential mechanisms of action”.

3 By a technical point of view, I don’t understand the choice to evaluate the antimicrobial activity of yeasts against spoilage fungi by mouth to mouth in vitro assay and then to use, in in vivo assays, the direct contact between yeasts and fungi inoculum. Why did the authors use these different approaches in the workflow? The control of fungi development on fruits, if linked with a VOCs mechanism, could be achieved i.e. with the exposure of fruits to volatiles released from GP plates inoculated with yeast strains and placed in closed boxes.

4 Finally, I strongly suggest to revising the introduction and discussion section with a deep bibliographic analysis related to VOCs produced by yeasts against fruit spoilage microorganisms, the different mechanisms of action of antagonistic yeasts (killer toxins, antimicrobial peptides, synergic activities among yeast volatiles, cell to cell contact and so on) and the cell targets (membrane, cell wall, DNA damage, proteins and so on). The English language use is comprehensive but requires minor edits.

Minor comments

L2-L3 Revise the title as suggested in major comments.

L36 Please deletes “because of spoilage” and replace with “caused by microbial spoilage”.

L41 Delete controlling substrates.

L91 This section is not necessary; it could be included in section 2.2. Remember to correct the numbering of other sections.

L129 …of an undesired fermentation from occurring… Delete from.

L134 and L227 Here and throughout the manuscript please do not use the numbering to cite a work in the text. It’s better i.e. “agar plate method as described by Núñez et al. (2015) [33].” “Mehlomakulu et al. (2014) reported only…against D. bruxellensis [28]”.

L137..spread-plated…

L157 Delete the space after …and 3 mm deep). After 15 minutes, 15… Delete the wound infliction.

L163-165 The biocontrol efficacy was evaluated by comparing the decay diameter of the negative control to those of the treated apples using the FRI. OK

Have the authors data related to the incidence of fungal growth (percentage of fungal growth on fruits on the total number of fruits inoculated) on apples and grapes? If yes, please include these data in the results.

L160-177 at -0.5°C for 4 weeks. Why the authors use this storage temperature for both apples and grapes? The reason could be included in the text.

L161-178 Delete ± and replace with ca.

L170 Delete inflicted…Replace with  ..were uniformly wounded (3 wound per spot)…

L183-184 Please delete “in order to inhibit fruit degradation”.

L195 Delete the dot in 2.9 mL.min-1.

L200-L203 Which kind of ANOVA was used? One way, two way…. Which kind of post-hoc test was used to separate the means?

L245 and L268 In my opinion, Figure 2A and 3A are not necessary. The photos could be included in supplementary material or under the figure 2B and 3B with two photos (control and treated plates) for each strain.

L298-L308 In the grape bioassay section there are no data related to fungal growth on grape berries (average diameter, radial inhibition, incidence and so on). 100% inhibition is not sufficient to describe the results obtained. Please include these data and the statistics as in the apple section.

L320 Check the spaces.

L326-328 Which is the mechanism of action of these VOCs against spoilage microorganisms?

L334-341 Please rewrite the conclusion section to match with the obtained results and the title of special issue. Future research on VOCs mechanism by antagonistic yeasts could be included in this section. Remember that in real conditions grapes are not stored in sealed jars and apples in tightly closed container. In a real packaging condition, interactions of yeast VOCs with VOCs produced by fruit tissue, and CO2 from respiration activity could play an important role in fungal growth suppression as well as in host defence response.

Author Response

REVIEWER 1

Comments and Suggestions for Authors

Manuscript number: 582883

Title: The Use of Candida pyralidae and Pichia kluyveri for  Control of Beverage and Fruit-spoilage Organisms

Major comments:

Comment: The manuscript titled “The Use of Candida pyralidae and Pichia kluyveri for  Control of Beverage and Fruit Spoilage Organisms” describes the use of antagonistic yeasts as biocontrol agents against food spoilage microorganisms such as fruit spoiling yeasts and fungi contaminating fruits. The manuscript has been sent for publication under the special issue “Microbial spoilage of beverages”. The authors describe the antimicrobial activity of these yeasts against fungi to control post-harvest spoilage of fresh fruits.

1 In my opinion, the objective of the work could be revised, matching with the title of the special issue. In particular, the objective could be the control of beverage spoilage microorganisms (yeasts) and fungi contaminating fruits used to produce beverages (fruit juices, smoothies) such as apple and grape described in the work. The term post-harvest and fungal pathogens could be deleted throughout the manuscript and replaced with “spoilage”, “spoilage fungi” of fruits “used for beverage production/processing”.

Response: As suggested, the objective has been revised

The most appropriate title could be “The Use of Candida pyralidae and Pichia kluyveri to Control Spoilage Microorganisms of Raw Fruits Used for Beverage Production”

Response The title has been revised

2 By a scientific point of view, the results are interesting for a broad audience since different target microorganisms are considered and different modes of action of Candida pyralidae and Pichia kluyveri such as killer toxin production, VOCs release, cell to cell contact and so on could be involved in the antimicrobial activity of these yeasts. The production of crude extracts using agro-food by-products such grape pomaces is interesting and matches with current literature (as recently reported for the production of killer toxins by recombinant yeasts using other food processing by-products). However, the conclusions should reflect the results obtained. For example, the sentence “Additionally, the cell suspensions of C. pyralidae and P. kluyverishowed growth inhibition properties against fungal growth (B. cinereaC. acutatum and R. stolonifer) through a VOCs mechanism” is correct only for in vitro assays but not for fruit challenge-tests. Indeed, the authors did not demonstrate that VOCs produced by yeasts are essentials for the antimicrobial activity detected on fruits. VOCs were detected on inoculum broths prepared from GP broth and not in the packaging headspace above the fruits; moreover, the proposed mechanism of action should require additional work related to evaluation of the antimicrobial activity of pure and/or combinations of VOCs and the definition of Minimum inhibitory concentrations of individual VOCs. In addition, inoculum procedure adopted (yeasts and fungi were simultaneously inoculated into wounded fruits) did not exclude other direct contact mechanisms such as the production and release of antifungal compounds by yeasts (antimicrobial peptides or other compounds inhibiting spore germination). For these reasons, I strongly suggest to be careful throughout the manuscript in the association between the antimicrobial activity detected and the mechanisms of action. They could be only suggested as “potential mechanisms of action”.

Response: The manuscript has been revised extensively and as suggested, the ‘mechanism of action has been extensively addressed Please refer to the manuscript.

3 By a technical point of view, I don’t understand the choice to evaluate the antimicrobial activity of yeasts against spoilage fungi by mouth to mouth in vitro assay and then to use, in in vivo assays, the direct contact between yeasts and fungi inoculum. Why did the authors use these different approaches in the workflow? The control of fungi development on fruits, if linked with a VOCs mechanism, could be achieved i.e. with the exposure of fruits to volatiles released from GP plates inoculated with yeast strains and placed in closed boxes.

Response: The choice of the evaluation techniques has been clarified in section 3.4.2 and materials and methods

4 Finally, I strongly suggest to revising the introduction and discussion section with a deep bibliographic analysis related to VOCs produced by yeasts against fruit spoilage microorganisms, the different mechanisms of action of antagonistic yeasts (killer toxins, antimicrobial peptides, synergic activities among yeast volatiles, cell to cell contact and so on) and the cell targets (membrane, cell wall, DNA damage, proteins and so on). The English language use is comprehensive but requires minor edits.

Response: The introduction and discussion section has been revised accordingly.

Minor comments

L2-L3 Revise the title as suggested in major comments.

Responses: The title has been revised

L36 Please deletes “because of spoilage” and replace with “caused by microbial spoilage”.

Responses: Deleted

L41 Delete controlling substrates.

Responses: Deleted

L91 This section is not necessary; it could be included in section 2.2. Remember to correct the numbering of other sections.

Responses: Addressed

L129 …of an undesired fermentation from occurring… Delete from.

Responses: Deleted

L134 and L227 Here and throughout the manuscript please do not use the numbering to cite a work in the text. It’s better i.e. “agar plate method as described by Núñez et al. (2015) [33].” “Mehlomakulu et al. (2014) reported only…against D. bruxellensis [28]”.

Responses: Corrected

L137..spread-plated…

Responses: Corrected

L157 Delete the space after …and 3 mm deep). After 15 minutes, 15… Delete the wound infliction.

Responses: Corrected

L163-165 The biocontrol efficacy was evaluated by comparing the decay diameter of the negative control to those of the treated apples using the FRI. OK

Have the authors data related to the incidence of fungal growth (percentage of fungal growth on fruits on the total number of fruits inoculated) on apples and grapes? If yes, please include these data in the results.

Response: Results are shown in Figure 4A. The average values per treatment (10 replicates) are shown, but only 3 fruits per treatments are shown.

L160-177 at -0.5°C for 4 weeks. Why the authors use this storage temperature for both apples and grapes? The reason could be included in the text.

Responses: The conditions have been clearly explained under section 2.4.1 and 2.4.2.

L161-178 Delete ± and replace with ca.

Responses: The reader may find the ca confusing in the figures. Therefore, for consistency, ± was adopted.

L170 Delete inflicted…Replace with  ..were uniformly wounded (3 wound per spot)…

Responses: Deleted and replaced

L183-184 Please delete “in order to inhibit fruit degradation”.

Responses: Deleted

L195 Delete the dot in 2.9 mL.min-1.

Responses: Deleted

L200-L203 Which kind of ANOVA was used? One way, two way…. Which kind of post-hoc test was used to separate the means?

Responses: Explained under section 2.6

L245 and L268 In my opinion, Figure 2A and 3A are not necessary. The photos could be included in supplementary material or under the figure 2B and 3B with two photos (control and treated plates) for each strain.

Responses: We fully agree. However, for the sake of the reader, we think these assay pictures can add value to the growth inhibition story.

L298-L308 In the grape bioassay section there are no data related to fungal growth on grape berries (average diameter, radial inhibition, incidence and so on). 100% inhibition is not sufficient to describe the results obtained. Please include these data and the statistics as in the apple section.

Responses: The explanation has been given. See section 3.4.2

L320 Check the spaces.

Responses: Checked

L326-328 Which is the mechanism of action of these VOCs against spoilage microorganisms?

Responses: The mechanisms have been explained in the manuscript

L334-341 Please rewrite the conclusion section to match with the obtained results and the title of special issue. Future research on VOCs mechanism by antagonistic yeasts could be included in this section. Remember that in real conditions grapes are not stored in sealed jars and apples in tightly closed container. In a real packaging condition, interactions of yeast VOCs with VOCs produced by fruit tissue, and COfrom respiration activity could play an important role in fungal growth suppression as well as in host defence response.

Responses: The conclusion section has been revised accordingly

Reviewer 2 Report

The study described explores the use of two yeast species as possible biocontrol agents and as source of biopreservation compounds. The manuscript is well structured and easy to read. Although the results obtained are quite interesting, the search for biocontrol agents is not that novel, though relevant within a sustainable and safe food production point of view.
However, the novelty should be clearly stated as well as the possible applications/bottlenecks/… Data about the inhibition assays should be carefully rechecked and standard deviations should be added to the bars in the graphs. More details on how certain choices were made, should be added. In addition, the matter should be addressed in a broader perspective, i.e. topics like regulatory implications, food-safety aspects (implications of biocontrol agents, also mycotoxins), sustainability, etc. should be reflected upon.

Below, in the different parts of the manuscript several aspects that need to be addressed are listed:

· Title:
the word “beverage” is not well defined. The title should be changed to “… control of fruit and fruit-derived beverage …”. Also, in the entire document, beverages should be changed to fruit-derived beverages.

· Abstract:
the part of the intro is too long compared to the results and conclusion part of the abstract. This should be changed. The novelty as well as the relevance of the results obtained need to be stated clearly.
L18: remove other preservatives
L20: remove the several …

· Introduction:

L35, L38, etc. fungal pathogens: do you mean phytopathogenic? And also, spoilage organisms are not necessary pathogenic or vice-versa. The terms “pathogenic” should be used carefully and if needed, adjusted or left out (to be checked throughout the entire manuscript). For example: “undesired fungal presence” instead of fungal pathogens on L35
Spoilage is used in the title, also on L39, 42, 254, etc.
L35: agricultural industry: is more than only this, is the entire agrifoodchain, please adjust
L38: why were B. cinerea, C. acutatum and R. stolonifer chosen? 2 references are mentioned. Botrytis cinerea occurs more on small fruits like strawberries and grapes. But for apples on the one hand and for grapes on the other hand, different species are more relevant and occurring. For instance, for apples Penicillium expansum is the most important species. This species also occurs in grapes, but not before harvest but during post-harvest storage. In addition, P. expansum is very important in view of food safety due to the production of mycotoxins, especially patulin! In the entire manuscript, the reflection to food safety due to mycotoxins is never addressed. This should be added!
This kind of studies also need to be conducted for other species like P. expansum in apples. If not conducted, this should be included as a prospective for further research in the conclusion section and in the abstract.

I refer to the book of Pitt & Hocking “Fungi and Food Spoilage”, more precisely p.385 and 388 for apples and grapes, respectively. There, other important references can be retrieved as well (e.g. Janisiewicz 1995 and 2000). In this book, there is also info on the natural yeasts present on this type of fruits; this should be added in the manuscript and reflected upon in relation to the possible real-life application of the yeast species tested in this study.
L39: “…to produce for about…”: what is meant by this?
L41-42: rephrase
L41: remove “controlling”
L52: “competing for nutrients and space with the other flora present”
L52: “ …; and the induction of host…”
L53: no capital for laminarinases
L56: ”… reducing fungal spore …”

· Mat&Meth:

Why only 1 control was taken into account (the one without yeast) for the in vitro experiment. If the 3 fungi would have been grown an a control medium, the influence of the composition of the medium used could have been seen. For instance, the influence of polyphenols from the grapes as possible antifungal compounds.
L67: remove “potential biocontrol”
L70: remove “and” after “Furthermore”
L71: 3 fungal species: give strain numbers and where they were isolated from (origin, type of commodity)
L76: info on pressing is lacking (only kPa is given); add info on details ((semi-)industrial, organic or conventional grapes (without fungicides applied during cultivation?), ….
L78: why 50 g/L was chosen? Mostly this concentration is higher in grape juice.
L81: why pH 5 was chosen? Mostly this concentration is lower in grape juice (below pH 4).
L82 and L137 and L159: replace “thereafter” by “subsequently”
L85: new line for the sentence/paragraph starting with “Yeasts were cultured …”
L88: obtain instead of attain
L91: why a separate title (2.3). Better just use new paragraph “For the inoculum preparation, …”
L92: full of the culture: which one (specify)
L100: rpm depends on type used, so should be expressed as “g” (grav. force – universal)
L101: “… before all mother...”
L106: “with” instead of “whereby”
L107: 8 superscript is not readable
L109-114: put chronologically (is confusing now for the reader), also remove non-relevant items (L111 “…which was …solidification”)
L116-131: mix of materials and methods and results – remove redundancy
L118: indeed higher concentration of sugar,  but why different than on L78? Also, which sugars were added?
L119: remove “respectively”
L122: throughout the entire manuscript, it is  very confusing for the reader how replicates are considered (technical ones, biological one?). Total number of samples, replicates, n=…; this is all used scattered and inconsistent according to me. Should be made more clear for the reader and made uniform for each experiment (text ánd tables). E.g. L156, L170, etc., legend figures, etc.
L123: rpm depends on type used, so should be expressed as “g” (grav. force – universal)
L123: remove “withdrawn”
Subtitles 2.5.1 and 2.5.2: remove “effect of …”; even better, rephrase and adjust according to the subtitle 2.4.1
L141: replace “whereby” by “with”
L141 and L142: representing
L144: same remark as above for subtitles (e.g. write VOCs-inhibition activity ‘VOCIA) as subtitle)
L152: add info on n, number of replicates in the material&methods section
L154: apple variety? Known to be very susceptible to the fungal species mentioned and of high economical relevance? Same comment for L170 (grapes)
L158: remove “then”
L162 and L177: containers and jars were sterilized prior to use? L177: remove “also”
L161 and L178: why no constant, stable temperature? How large was the deviation?
L166: incorrect terminology “maintenance”
L173: how was this checked (until the dried wounds were filled…)? Why not submerged in solution?
L183: “evolution”: incorrect terminology
L184: inhibit fruit degradation: I do not understand. Would this occur after such short time?
L184: -d8 subscript for 8
L185 and L197: give in full SPME and MSD
L192: details of Thermo (country, …) are lacking
L201: give more details on the analysis of variance test performed

· Results & discussion:

L206: show
L208: D. anomala is not depicted in Fig 1, so remove from text. On L230 “no data is mentioned”.
L215: “reasonable” is subjective… What do you mean by this?
L215-216: “… who successfully carried out …”
L220-222: is not clear; rephrase
L223: substrate concentration?
L226: R. stolonifera is a fungus – so belongs in discussion of 3.2
L229: a2)) (bracket missing)
L242-244: is part of in vivo, not of 3.2
FIGURE 1c: but which one is Candida/Pichia?
FIGURE 2A: lay-out insufficient and also to wide
FIGURE 2B and 3B: not relevant, can be removed and info added in the text in 1 single sentence. Also, all results (%) are the same … seems very strange. In fact, for Fig 2, the radial-inhibition assay was conducted as described in 2.5.1 (FRI). It seems very unlikely that all FRI-values for the 3 different fungi considered would be 100%. However, indeed, if no growth was observed, as seems to be the case, the FRI will be 100% for all three tested. But in that case, mentioning this in the text is advised, and not an extra figure without any additional info. Also, why n=1 in fig2B and n=10 in fig3B?
L255-256: is this the only novel thing?
L255: what is meant with “refined media”?
L258: “produced” instead of “producing”
L258: “…VOCs on fungal growth on fruit ….”
L266: “additionally” instead of “further”
L265: Fusarium first time full
L277 versus L298: Latin name for grapes as well
L279: “considerable” = statistically significant?
L282: 100% fungal-growth inhibition? But what about e1, e2?
L283-284: split into two sentences. “C. pyralidae showed a 43% and …”
L288: “Although…”? I do not understand the meaning of this sentence. Also, “higher” instead of “greater” (but compared to what?)
L291: Fig4A: for R. stolonifer item f2 shows no infection but f1 and f3 do. Is there an explanation for this?
L299-300: remove this sentence
L315: “Identification of” (remove The) and species not italic if a sentence is already put in italic
L317: 25 VOCs for both yeasts? Exactly the same ones and both exactly 25?
L317-322: no capitals for the different compounds
L324: “…have indeed been linked …”
L326: “Some”= important? Safe? Commonly used?
Entire paragraph: remove “-“ between in-vivo and in-vitro and put italic: in vivo and in vitro

· Conclusions:

topics like regulatory implications, food-safety aspects, sustainability, etc. should be reflected upon

L: 336 “broader” instead of “higher”? and compared to what?

· References: make this more uniform (e.g. no capitals in article titles (ref number 2), no “;” at the end of ref number 5, italic “in vitro” and “in situ” for ref number 18, capital Penicillia for ref number 33

In addition, following aspects need to be addressed as well:

· The need for more details on how certain choices were made:

Choice of fungal spoilage species (see remark above)

· English writing – grammar:

-  The use of hyphens in English language should be consulted and applied

L58/68/… beverage-spoilage organisms, L59 fruit-spoilage organism, L63/71 South-African fruit etc. (throughout the entire manuscript), L70/71 spoilage- and disease-causing agents, L74 spore-suspension preparation, L87: scrapping off (without), L97 fungal-spore suspensions, L104 (and other L105/129…) growth-inhibition assay, L132 post-harvest, L152 in vivo, L161 shelf-life conditions, etc.

- Singular versus plural: many verbs do not correspond to the subject (e.g. should be changed on L15 poses, L19 is, L55 mechanisms, 102 solutions were, L106 were, L108 was, L111 yeasts, L120 were, L139 was, L162 was, L182 was, L286 have), L66 selection, etc.), L254 yeasts

- Italic or not: L 132, 133, 152, 163, 276, 277, … species names; in vivo

· Inconsistencies of notation/punctuation:

- n=…. Versus replicates

- temperatures: comma before the symbol or not, and also ° in superscript – this is not always written in the same way and should be adjusted throughout the entire manuscript (L83/86/…)

- remove 1 interval on L79 before 0.045, L94 before C., L102

- L166: add an interval before “µl”

- L158, L173: remove 1x (not used elsewhere)

- L158: remove “.” before mL-1

- L197: remove “.” before min

- L215: name author is mentioned, but is not the case for other references in the text (see mat&meth section and also line 227 [28]) – to be checked and adjusted according to the instructions for authors

· Figures:

- Lay-out: the width is often not adequate and should be changed

- Leave out Fig2B and Fig3B

- Legend: adjust according to experiments described (e.g. n= …), and be consistent whether or not to mention the full genus name or not (e.g. Pichia or P. L236

Author Response

REVIEWER 2

Comments and Suggestions for Authors

The study described explores the use of two yeast species as possible biocontrol agents and as source of biopreservation compounds. The manuscript is well structured and easy to read. Although the results obtained are quite interesting, the search for biocontrol agents is not that novel, though relevant within a sustainable and safe food production point of view. 
However, the novelty should be clearly stated as well as the possible applications/bottlenecks/… Data about the inhibition assays should be carefully rechecked and standard deviations should be added to the bars in the graphs. More details on how certain choices were made, should be added. In addition, the matter should be addressed in a broader perspective, i.e. topics like regulatory implications, food-safety aspects (implications of biocontrol agents, also mycotoxins), sustainability, etc. should be reflected upon.

Responses: Thank You very much for this valuable input. As suggested, the manuscript has been revised accordingly

Below, in the different parts of the manuscript several aspects that need to be addressed are listed:

Title
the word “beverage” is not well defined. The title should be changed to “… control of fruit and fruit-derived beverage …”. Also, in the entire document, beverages should be changed to fruit-derived beverages.

Responses: Title changed

Abstract
the part of the intro is too long compared to the results and conclusion part of the abstract. This should be changed. The novelty as well as the relevance of the results obtained need to be stated clearly.

Responses: Noted and addressed
L18: remove other preservatives

Responses: Removed

L20: remove the several …

Responses: Removed

Introduction:

L35, L38, etc. fungal pathogens: do you mean phytopathogenic? And also, spoilage organisms are not necessary pathogenic or vice-versa. The terms “pathogenic” should be used carefully and if needed, adjusted or left out (to be checked throughout the entire manuscript). For example: “undesired fungal presence” instead of fungal pathogens on L35
Spoilage is used in the title, also on L39, 42, 254, etc.

Responses: Corrected

L35: agricultural industry: is more than only this, is the entire agrifoodchain, please adjust

Responses: Adjusted

L38: why were B. cinerea, C. acutatum and R. stolonifer chosen? 2 references are mentioned. Botrytis cinerea occurs more on small fruits like strawberries and grapes. But for apples on the one hand and for grapes on the other hand, different species are more relevant and occurring. For instance, for apples Penicillium expansum is the most important species. This species also occurs in grapes, but not before harvest but during post-harvest storage. In addition, P. expansum is very important in view of food safety due to the production of mycotoxins, especially patulin! In the entire manuscript, the reflection to food safety due to mycotoxins is never addressed. This should be added!
This kind of studies also need to be conducted for other species like P. expansum in apples. If not conducted, this should be included as a prospective for further research in the conclusion section and in the abstract.

Response: Changed and added more information.

I refer to the book of Pitt & Hocking “Fungi and Food Spoilage”, more precisely p.385 and 388 for apples and grapes, respectively. There, other important references can be retrieved as well (e.g. Janisiewicz 1995 and 2000). In this book, there is also info on the natural yeasts present on this type of fruits; this should be added in the manuscript and reflected upon in relation to the possible real-life application of the yeast species tested in this study.

Responses: The manuscript has been revised extensively and further related information has been added to the manuscript

L39: “…to produce for about…”: what is meant by this?

Responses: Corrected

L41-42: rephrase

Responses: Corrected

L41: remove “controlling”

Responses: Corrected

L52: “competing for nutrients and space with the other flora present”

Responses: Corrected

L52: “ …; and the induction of host…”

Responses: Corrected

L53: no capital for laminarinases

Responses: Corrected

 L56: ”… reducing fungal spore …”

Responses: Corrected

Mat&Meth:

Why only 1 control was taken into account (the one without yeast) for the in vitro experiment. If the 3 fungi would have been grown an a control medium, the influence of the composition of the medium used could have been seen. For instance, the influence of polyphenols from the grapes as possible antifungal compounds.

Responses: More information regarding this has been provided in the manuscript.

L67: remove “potential biocontrol”

Responses: Corrected

L70: remove “and” after “Furthermore”

Responses: Corrected

L71: 3 fungal species: give strain numbers and where they were isolated from (origin, type of commodity)

Responses: Corrected

L76: info on pressing is lacking (only kPa is given); add info on details ((semi-)industrial, organic or conventional

grapes (without fungicides applied during cultivation?), ….

Responses: Clarified

L78: why 50 g/L was chosen? Mostly this concentration is higher in grape juice.

Responses: Clarified. In a previous study, refined media were used. Usually these refined media contain a glucose concentration of 10-20 g/L. The normal grape pomace extract is diluted to get 50 g/L, which contains more sugar, but is a cheaper medium than any of the normal refined media available. One of the objectives of this project is to find a cheap media that can be used for cultivation of microorganisms.

L81: why pH 5 was chosen? Mostly this concentration is lower in grape juice (below pH 4).

Response:  In a previous study, the optimal pH for biocontrol compound production was shown to be 5

L82 and L137 and L159: replace “thereafter” by “subsequently”

Responses: Corrected

L85: new line for the sentence/paragraph starting with “Yeasts were cultured …”

Responses: Corrected

L88: obtain instead of attain

Responses: Corrected

L91: why a separate title (2.3). Better just use new paragraph “For the inoculum preparation, …”

Responses: Corrected

L92: full of the culture: which one (specify)

Responses: Corrected

L100: rpm depends on type used, so should be expressed as “g” (grav. force – universal)

Responses: For consistency with other similar works published, ‘rpm’ units were adopted.

L101: “… before all mother...”

Responses: Corrected

L106: “with” instead of “whereby”

Responses: Corrected

L107: 8 superscript is not readable

Responses: Corrected

L109-114: put chronologically (is confusing now for the reader), also remove non-relevant items (L111 “…which was …solidification”)

Responses: Corrected

L116-131: mix of materials and methods and results – remove redundancy

Responses: Redundancy has been eliminated

L118: indeed higher concentration of sugar,  but why different than on L78? Also, which sugars were added?

Responses: Clarified

L119: remove “respectively”

Responses: Corrected

L122: throughout the entire manuscript, it is  very confusing for the reader how replicates are considered (technical ones, biological one?). Total number of samples, replicates, n=…; this is all used scattered and inconsistent according to me. Should be made more clear for the reader and made uniform for each experiment (text ánd tables). E.g. L156, L170, etc., legend figures, etc.

Responses: The issue of the samples and replicates has been addressed. Please refer to the results and materials and methods.

L123: rpm depends on type used, so should be expressed as “g” (grav. force – universal)

Responses: Please see above

L123: remove “withdrawn”

Responses: Corrected

Subtitles 2.5.1 and 2.5.2: remove “effect of …”; even better, rephrase and adjust according to the subtitle 2.4.1

Responses: Corrected

L141: replace “whereby” by “with”

Responses: Corrected

L141 and L142: representing

Responses: Corrected

L144: same remark as above for subtitles (e.g. write VOCs-inhibition activity ‘VOCIA) as subtitle)

Responses: Corrected

152: add info on n, number of replicates in the material&methods section

Responses: Added

L154: apple variety? Known to be very susceptible to the fungal species mentioned and of high economical relevance? Same comment for L170 (grapes)

Responses: Corrected

L158: remove “then”

Responses: Corrected

L162 and L177: containers and jars were sterilized prior to use? L177: remove “also”

Responses: Corrected

L161 and L178: why no constant, stable temperature? How large was the deviation?

Responses: Clarified

L166: incorrect terminology “maintenance”

Responses: Corrected

L173: how was this checked (until the dried wounds were filled…)? Why not submerged in solution?

Responses: Clarified

L183: “evolution”: incorrect terminology

Responses: Corrected

L184: inhibit fruit degradation: I do not understand. Would this occur after such short time?

Responses: Clarified

L184: -d8 subscript for 8

Responses: Corrected

L185 and L197: give in full SPME and MSD

Responses: Corrected

L192: details of Thermo (country, …) are lacking

Responses: Corrected

L201: give more details on the analysis of variance test performed

Responses: Corrected

Results & discussion:

L206: show

Responses: Corrected

L208: D. anomala is not depicted in Fig 1, so remove from text. On L230 “no data is mentioned”.

Responses: Corrected

L215: “reasonable” is subjective… What do you mean by this?

Responses: Clarified

L215-216: “… who successfully carried out …”

Responses: Corrected

L220-222: is not clear; rephrase

Responses: Corrected

L223: substrate concentration?

Responses: Clarified

L226: R. stolonifera is a fungus – so belongs in discussion of 3.2

Responses: Corrected

L229: a2)) (bracket missing)

Responses: Corrected

L242-244: is part of in vivo, not of 3.2

Responses: Corrected

FIGURE 1c: but which one is Candida/Pichia?

Responses: Corrected

FIGURE 2A: lay-out insufficient and also to wide

Responses: Corrected

FIGURE 2B and 3B: not relevant, can be removed and info added in the text in 1 single sentence. Also, all results (%) are the same … seems very strange. In fact, for Fig 2, the radial-inhibition assay was conducted as described in 2.5.1 (FRI). It seems very unlikely that all FRI-values for the 3 different fungi considered would be 100%. However, indeed, if no growth was observed, as seems to be the case, the FRI will be 100% for all three tested. But in that case, mentioning this in the text is advised, and not an extra figure without any additional info. Also, why n=1 in fig2B and n=10 in fig3B?

Responses: The importance of the 2 figures has been emphasised in the manuscript and mistakes corrected.

L255-256: is this the only novel thing?

Response: No. Clarified

L255: what is meant with “refined media”?

Responses: Clarified

L258: “produced” instead of “producing”

Responses: Corrected

L258: “…VOCs on fungal growth on fruit ….”

Responses: Corrected

L266: “additionally” instead of “further”

Responses: Corrected

L265: Fusarium first time full

Responses: Corrected

L277 versus L298: Latin name for grapes as well

Responses: Corrected

L279: “considerable” = statistically significant?

Responses: Clarified

L282: 100% fungal-growth inhibition? But what about e1, e2?

Responses: Clarified

L283-284: split into two sentences. “C. pyralidae showed a 43% and …”

Responses: Corrected

L288: “Although…”? I do not understand the meaning of this sentence. Also, “higher” instead of “greater” (but compared to what?)

Responses: Clarified

L291: Fig4A: for R. stolonifer item f2 shows no infection but f1 and f3 do. Is there an explanation for this?

Responses: Clarified

L299-300: remove this sentence

Responses: Corrected

L315: “Identification of” (remove The) and species not italic if a sentence is already put in italic

Responses: Corrected

L317: 25 VOCs for both yeasts? Exactly the same ones and both exactly 25?

Responses: Yes for both of them

L317-322: no capitals for the different compounds

Responses: Corrected

L324: “…have indeed been linked …”

Responses: Corrected

L326: “Some”= important? Safe? Commonly used?

Responses: Supposedly because they have not been reported to be harmful to consumers

Entire paragraph: remove “-“ between in-vivo and in-vitro and put italic: in vivo and in vitro

Responses: Corrected

Conclusions:

topics like regulatory implications, food-safety aspects, sustainability, etc. should be reflected upon

Responses: Corrected

L: 336 “broader” instead of “higher”? and compared to what?

Responses: Corrected

References: make this more uniform (e.g. no capitals in article titles (ref number 2), no “;” at the end of ref number 5, italic “in vitro” and “in situ” for ref number 18, capital Penicilliafor ref number 33

Responses: Corrected

In addition, following aspects need to be addressed as well:

The need for more detailson how certain choiceswere made: Choice of fungal spoilage species (see remark above)

Responses: Corrected above

Englishwriting – grammar:

Responses: Checked

The use of hyphens in English language should be consulted and applied

L58/68/… beverage-spoilage organisms, L59 fruit-spoilage organism, L63/71 South-African fruit etc. (throughout the entire manuscript), L70/71 spoilage- and disease-causing agents, L74 spore-suspension preparation, L87: scrapping off (without), L97 fungal-spore suspensions, L104 (and other L105/129…) growth-inhibition assay, L132 post-harvest, L152 in vivo, L161 shelf-life conditions, etc.

Responses: Checked and corrected

- Singular versus plural: many verbs do not correspond to the subject (e.g. should be changed on L15 poses, L19 is, L55 mechanisms, 102 solutions were, L106 were, L108 was, L111 yeasts, L120 were, L139 was, L162 was, L182 was, L286 have), L66 selection, etc.), L254 yeasts

Responses: Checked and corrected

- Italic or not: L 132, 133, 152, 163, 276, 277, … species names; in vivo

Responses: Checked and corrected

Inconsistencies of notation/punctuation:

- n=…. Versus replicates

Responses: Checked and corrected

- temperatures: comma before the symbol or not, and also ° in superscript – this is not always written in the same way and should be adjusted throughout the entire manuscript (L83/86/…)

Responses: Checked and corrected

- remove 1 interval on L79 before 0.045, L94 before C., L102

Responses: Checked and corrected

- L166: add an interval before “µl”

Responses: Checked and corrected

- L158, L173: remove 1x (not used elsewhere)

Responses: Checked and corrected

- L158: remove “.” before mL-1

Responses: Checked and corrected

- L197: remove “.” before min

Responses: Checked and corrected

- L215: name author is mentioned, but is not the case for other references in the text (see mat&meth section and also line 227 [28]) – to be checked and adjusted according to the instructions for authors

Responses: Checked and corrected

Figures:

- Lay-out: the width is often not adequate and should be changed

Responses: Checked

- Leave out Fig2B and Fig3B

Responses: Clarified in the text

- Legend: adjust according to experiments described (e.g. n= …), and be consistent whether or not to mention the full genus name or not (e.g. Pichiaor P. L236

Responses: Checked and corrected

Round 2

Reviewer 1 Report

Manuscript number: 582883

Title: The Use of Candida pyralidae and Pichia kluyveri for Control of Beverage and Fruit-spoilage Organisms

The authors considered large part of the issues highlighted by the reviewer in the revised version. However, some aspects were not fully addressed.

- In particular, figures were not changed. In the figure 4B deviation standard for each bar is lacking. Is not sufficient to report the deviation standard in the figure 4A!! These values have to be included in Figure 4B. Please correct

- The information provided by authors explained the number of replicates per treatment. In the grape bioassay 10 replicates each composed by 10 grape berries were used. However, I did not find in the Figure 5 any data related to the number of spoiled berries per treatment. Please include the mean value of spoiled berries with the related standard deviation (i.e. in the case of 100% inhibition control berries = 100 ± 0 and treated berries 0 ± 0).

- A deep bibliographic analysis related to VOCs produced by yeasts and their mechanism of action (in particular the cell targets, i.e. cell wall, DNA, mitochondria, proteins) was not considered since the number of references did not increase. I strongly suggest adding results related to these aspects in the introduction and discussion sections. The following papers are examples that could be considered: doi.org/10.1016/j.tifs.2015.11.003, , doi.org/10.1016/j.fm.2019.01.008, doi.org/10.1016/j.biocontrol.2018.05.014, doi.org/10.1016/j.postharvbio.2019.03.002. Other papers such as the reference 27 (doi.org/10.1016/j.fm.2016.09.005) could be further analysed and discussed.

The reviewer is not principal investigator or co-author of these papers, so the aim of this request is not to have more citations but to place the submitted paper in a broad scientific context.

Minor edits:

L305 Ten replicates consisting of 10 grapes per jar were tested. Please replace grapes with grape berries

Author Response

Reviewer 1

Manuscript number: 582883

Title: The Use of Candida pyralidae and Pichia kluyveri for Control of Beverage and Fruit-spoilage Organisms

The authors considered large part of the issues highlighted by the reviewer in the revised version. However, some aspects were not fully addressed.

Point 1: - In particular, figures were not changed. In the figure 4B deviation standard for each bar is lacking. Is not sufficient to report the deviation standard in the figure 4A!! These values have to be included in Figure 4B. Please correct.

Answer:

The standard deviations have been added to figure 4 B.

Point 2: - The information provided by authors explained the number of replicates per treatment. In the grape bioassay 10 replicates each composed by 10 grape berries were used. However, we did not find in the Figure 5 any data related to the number of spoiled berries per treatment. Please include the mean value of spoiled berries with the related standard deviation (i.e. in the case of 100% inhibition control berries = 100 ± 0 and treated berries 0 ± 0).

Answer: the information regarding the number of berries spoiled in the negative control jars and the unspoiled berries in the treated jars have now been clarified, accompanied by the related mean values and standard deviations.

Point 3: - A deep bibliographic analysis related to VOCs produced by yeasts and their mechanism of action (in particular the cell targets, i.e. cell wall, DNA, mitochondria, proteins) was not considered since the number of references did not increase. I strongly suggest adding results related to these aspects in the introduction and discussion sections. The following papers are examples that could be considered: doi.org/10.1016/j.tifs.2015.11.003, , doi.org/10.1016/j.fm.2019.01.008, doi.org/10.1016/j.biocontrol.2018.05.014, doi.org/10.1016/j.postharvbio.2019.03.002. Other papers such as the reference 27 (doi.org/10.1016/j.fm.2016.09.005) could be further analysed and discussed.

Answer: Thank you for suggesting some published work to address this valuable comment. This has been addressed accordingly.

Point 4: The reviewer is not principal investigator or co-author of these papers, so the aim of this request is not to have more citations but to place the submitted paper in a broad scientific context.

Answer: We sincerely acknowledge the reviewer contribution and we are grateful for the will to place the paper in a broad scientific context. 

Minor edits:

Point 5: L305 Ten replicates consisting of 10 grapes per jar were tested. Please replace grapes with grape berries

Answer:

This suggestion has been implemented

Reviewer 2 Report

In the revised manuscript, most of the comments/suggestions have been addressed and taken into account. Some aspects, however, could not be found in the revised manuscript:

abstract: novelty/relevance of the study is still not clearly described in the abstract (1 sentence should be added)
introduction: HACCPs should be mentioned in 1 or 2 sentences (pre/postharvest, processing, storage) in order to get the full picture of the problem.
In fact, I believe this is the main comment on the manuscript, namely the lack of a clear problem-describing paragraph including ALL aspects (spoilage, food-safety, economic losses), and the relevance/meaning of the results presented in order to solve/address these issues.This is either scattered in the manuscript or not addressed. One concise paragraph could solve this. L38-39: sentence seems to be standing alone and not really matching the rest. Please rephrase. 25 VOCs: all the same for both yeasts, so add this info to the manuscript. indeed, some more info on the problem on fungi in apples, grapes is added, but still, Penicillium expansum (and related to this, the patulin issue), the biggest issue for apples (and PAT has regulatory limits!!!) is not mentioned as suggested previously. Although this was not the focus of the study, this should be addressed (e.g. further research). In fact, these are not mentioned in the entire manuscripts. discussion/conclusion: nothing on the food-safety implications (e.g. how will this (positively?) effect the mycotoxin burden in fruit-based beverages? What about implications for fermentation of fruit-based beverages (e.g. cider)?Some practical considerations of possible implementation should be addressed. FIGURES: these are still not ok. To wide; lay-out should be improved for all! It is still not clear what the added value/info of Fig. 2B is. Also, L230: "similar", but in fact, they are all the same (100%).

Some minor aspects to address are:

L18: was assessed ... L24: remove "the" L83: g L-1 (not g/L) L88: growth-inhibition L109: missing bracket ")" L114: growth-inhibition L129: add "(VOCs)" in title L203 and L204: et al with or without "." after "al"; elsewhere in the manuscript, it is written without "."

Author Response

Reviewer 2

In the revised manuscript, most of the comments/suggestions have been addressed and taken into account. Some aspects, however, could not be found in the revised manuscript:

Point 1: abstract: novelty/relevance of the study is still not clearly described in the abstract (1 sentence should be added)

Answer:

Thank you for the comment and suggestion, a sentence highlighting the novelty of this work has been added

 Point 2: introduction: HACCPs should be mentioned in 1 or 2 sentences (pre/postharvest, processing, storage) in order to get the full picture of the problem. In fact, I believe this is the main comment on the manuscript, namely the lack of a clear problem-describing paragraph including ALL aspects (spoilage, food-safety, economic losses), and the relevance/meaning of the results presented in order to solve/address these issues.This is either scattered in the manuscript or not addressed. One concise paragraph could solve this.

Answer: The HACCP aspect with regard to the different stages at which fruits and beverages contamination could occur has been added, accompanied by the related references. The meaning of the current results in addressing the problem raised has been addressed

 Point 3: L38-39: sentence seems to be standing alone and not really matching the rest. Please rephrase.

Answer:

Sentence removed

Point 4: 25 VOCs: all the same for both yeasts, so add this info to the manuscript. indeed, some more info on the problem on fungi in apples, grapes is added, but still, Penicillium expansum (and related to this, the patulin issue), the biggest issue for apples (and PAT has regulatory limits!!!) is not mentioned as suggested previously. Although this was not the focus of the study, this should be addressed (e.g. further research). In fact, these are not mentioned in the entire manuscripts. discussion/conclusion: nothing on the food-safety implications (e.g. how will this (positively?) effect the mycotoxin burden in fruit-based beverages? What about implications for fermentation of fruit-based beverages (e.g. cider) ?Some practical considerations of possible implementation should be addressed.

Answer:

 The information clarifying that the 25 VOCs were the same for both yeasts has been added. Also, the information on the “patulin” as suggested, has been added and discussed

Point 5: FIGURES: these are still not ok. To wide; lay-out should be improved for all! It is still not clear what the added value/info of Fig. 2B is. Also, L230: "similar", but in fact, they are all the same (100%).

Answer:

The quality of the figures has been improved and the similarity of the profiles was clarified.

Some minor aspects to address are:

Point 6: L18: was assessed ... L24: remove "the" L83: g L-1 (not g/L) L88: growth-inhibition L109: missing bracket ")" L114: growth-inhibition L129: add "(VOCs)" in title L203 and L204: et al with or without "." after "al"; elsewhere in the manuscript, it is written without "."

Answer:

All the above minor edits have been implemented